# Reliable Hallmarks and Biomarkers of Senescent Lymphocytes

**DOI:** 10.3390/ijms242115653

**Published:** 2023-10-27

**Authors:** Yuliya S. Martyshkina, Valeriy P. Tereshchenko, Daria A. Bogdanova, Stanislav A. Rybtsov

**Affiliations:** 1Division of Immunobiology and Biomedicine, Center for Genetics and Life Sciences, Sirius University of Science and Technology, Olimpiyskiy Ave. b.1, Sirius 354340, Krasnodar Region, Russia; martyshkina.yi@learn.siriusuniversity.ru (Y.S.M.);; 2Resource Center for Cell Technology and Immunology, Sirius University of Science and Technology, Olimpiyskiy Ave. b.1, Sirius 354340, Krasnodar Region, Russia

**Keywords:** senescent lymphocytes, immunosenescence, biomarkers of senescence

## Abstract

The phenomenon of accumulation of senescent adaptive immunity cells in the elderly is attracting attention due to the increasing risk of global epidemics and aging of the global population. Elderly people are predisposed to various infectious and age-related diseases and are at higher risk of vaccination failure. The accumulation of senescent cells increases age-related background inflammation, “Inflammaging”, causing lymphocyte exhaustion and cardiovascular, neurodegenerative, autoimmune and cancer diseases. Here, we present a comprehensive contemporary review of the mechanisms and phenotype of senescence in the adaptive immune system. Although modern research has not yet identified specific markers of aging lymphocytes, several sets of markers facilitate the separation of the aging population based on normal memory and exhausted cells for further genetic and functional analysis. The reasons for the higher predisposition of CD8^+^ T-lymphocytes to senescence compared to the CD4^+^ population are also discussed. We point out approaches for senescent-lymphocyte-targeting markers using small molecules (senolytics), antibodies and immunization against senescent cells. The suppression of immune senescence is the most relevant area of research aimed at developing anti-aging and anti-cancer therapy for prolonging the lifespan of the global population.

## 1. Introduction

Improved access to a modern healthcare system has significantly increased life expectancy and, as a result, the proportion of the elderly population has risen in both developing and developed countries. According to UN estimates, by 2050, the proportion of people aged 65 and over will increase from 10 to 16 percent of the global population [1].

Governments of different countries, in order to reduce public spending, have increased the retirement age with the subsequent aging of the working population. This rapid increase in life expectancy is, unfortunately, not accompanied by healthy aging; on the contrary, there is an increase in the diseases associated with age and the proportion of disabled citizens, which further puts the burden on the healthcare system [2,3].

To combat age-related diseases, morbidity and frailty, it is necessary to combine the efforts of society, science and new innovative approaches in anti-aging medicine and regenerative technologies to introduce anti-senescence therapy, the treatment of metabolic and inflammatory diseases. These efforts will definitely bridge the gap between lifespan and healthspan for future economic prosperity and equitable global wellbeing.

Aging also affects the immune system, reducing the effectiveness of immune surveillance and resulting in imbalanced immune responses, autoimmune disorders and anemia. Recently, it was documented that the accumulation of senescent cells in all immune populations is a hallmark of immune aging and an accelerator of inflammatory aging and immune dysfunction in older adults. Immunosenescent cells persist in a state of cell cycle arrest and cause a number of negative and destructive effects, such as a decrease in the ability to control malignancy and cleanse tissues of senescent cells, causing their accumulation in the entire body [4]; stem cell senescence [5]; the decreased efficiency of immune response to vaccination [6,7]; and increased susceptibility to various infectious diseases [8].

The immunosenescent state impedes T-lymphocytes from successfully fighting tumor cells, which shows its pivotal role in cancer development [9]. The role of immunosenescent cells has been shown in the development of diabetes [10] and neurodegenerative [11,12], autoimmune [13,14,15] and cardiovascular diseases [16] in old age. Immunosenescence also likely contributes to an increased susceptibility to a severe form of coronavirus infection and increases morbidity and mortality from infectious diseases among elderly individuals [17].

The study of the relationship between the aging of the immune system and age-associated diseases is becoming one of the most relevant areas of current immunological research for the development of new effective approaches for the prevention and treatment of age-related pathologies. Basic research of the human immune system requires knowledge of relevant and specific markers of senescent immune cells that allow unambiguous determination of their senescent status, malignant properties and secretory and metabolic phenotype. Recent studies of lymphocyte populations in individuals of different ages have also revealed the fine regulation of the immune response associated with disruption of signaling interactions and features of homeostatic proliferation [18,19,20,21].

The most important studies of the role of senescent lymphocytes in the aging of the body and their specific markers and properties published in recent years require a systematic discussion. Thus, our review is devoted to a comprehensive analysis of the features of aged and senescent lymphocytes, identification of biomarkers and determination of targets of specific senescent populations.

## 2. The Hallmarks of Senescent Immune Cells

### 2.1. Morphological Changes

Among the distinct features of senescent cells, the most noticeable external features are flattened morphology and increased size, which are associated, among other factors, with reduced expression of scaffold proteins such as caveolin-1, Rac1, and CDC42 [22]. The appearance of enlarged and flattened cells was detected after prolonged exposure of the B-cell lymphoma line A20 to sodium arsenite [23]. Along with decreased functionality, changes in the shape and size of immune cells during aging are probably one of the reasons for the age-related disruptions in the architectural organization and structure observed in tissues such as lymph nodes [24] and the thymus [25].

Significant changes affect many cellular organelles with senescence: the number and size of vacuoles, cytoplasmic filaments and nucleus and nucleolus enlargement (although an increase in the ratio of nucleus to cytoplasm also characterizes young cells such as naive lymphocytes and stem cells). Sometimes, senescent cells may be multinucleated and contain increased numbers of lysosomes and Golgi complexes [26]. In addition, the mitochondria in lymphocytes of people over 60 are enlarged and irregularly spaced, and their cristae are replaced by a myelin-like structure (or other electron-dense substance) [27].

The lysosomes of senescent cells accumulate lipofuscin, a yellow-brown “age pigment”, which is an undegradable aggregate of oxidized lipids, covalently cross-linked proteins, oligosaccharides and transition metals, formed by iron-catalyzed oxidation and polymerization of different cellular structures and macromolecules [28,29]. Lipofuscin granules and tubuloreticular inclusions (TRI) were found in the cytoplasm of lymphocytes from donors older than 60 years, and their content in cells increases with age [27]. Significant accumulation of lipofuscin was also observed by Gerland L. M. et al. during the replicative senescence of lymphocytes in vitro [30]. Due to its autofluorescence, lipofuscin can be easily visualized via fluorescence microscopy [31,32] and flow cytometry [33] as well as by histochemical techniques such as Sudan Black B staining, making this biomarker useful for detection of senescent cells [34]. The accumulation of lipofuscin during aging correlates well with other established markers of cell senescence, such as increased activity of senescent-associated beta-galactosidase (SA-β-Gal), lack of proliferation (determined by the absence of the Ki-67 marker) [35] and augmented expression of p16 and p21 [36].

### 2.2. Surface Markers

Although no universal surface marker of senescence has been found for all cell types, markers specific to individual cell populations that correlate well with the senescence of these cells have been identified [37]. Loss of the CD28 costimulatory molecule expression, which plays an important role in T-cell activation, was one of the first distinguishing features proposed to identify senescent cells [38,39]. These early studies found that during replicative senescence, the decrease in CD28 expression is more pronounced in a population of CD8+ T-lymphocytes (CD8+ CTLs) in comparison with CD4+ T-cells (CD4+ Ths) [39]. Loss of CD27, another costimulatory molecule, occurs as a response to chronic antigenic load and is also considered a marker of senescent cells [40]. Increased numbers of CD28- and CD27- T-cells are observed in the elderly [41,42]. As CD27 and CD28 are important co-stimulatory molecules essential for TCR signal transduction and T-cell priming, their loss by senescent T-cells is in agreement with the loss of their function and ability to activate. Naive CD28+ T-cells exist transiently and mature rapidly upon stress or antigen recognition. This could explain why no significant number of senescent cells were found among CD28+ T-cells. In contrast, long-term persistent memory cells are more likely to undergo replicative and metabolic stress or DNA damage, resulting in a switch to a senescent state. Indeed, memory cell markers such as CD45R0 have been found on senescent cells [43]. However, unlike memory cells, senescent cells demonstrate restoration of the expression of the naïve lymphocyte marker CD45RA, while remaining CD27-negative, in response to antigenic stimulation [40].

Expression of the mature NK cell receptor CD57 correlates with the loss of cell proliferation ability and shorter telomere length [44]. Upregulation of KLRG1 (killer-cell lectin like receptor G1) increases dramatically with age, mainly in the CD8+ CTL population and correlates with loss of T-cell proliferation capacity. Following this observation, KLRG1 has also been proposed as a marker of senescent cells and exhaustion of lymphocytes [45,46].

Senescent NK cells (natural killers) and T-lymphocytes also express TIGIT (immunoreceptor for T-cells with Ig and ITIM domains) on their surface. The number of TIGIT+ cells correlates with age, especially strongly in the CD8+ T-lymphocyte population [47]. However, these surface markers do not accurately define senescent immune cells, since they also characterize the process of cell differentiation and overlap with the phenotype of memory cells [48,49]. Moreover, cells that have lost CD28 expression and/or exhibit various levels of expression of CD57 and other senescence markers described above retain the ability to proliferate, making it difficult to classify these cells as true senescent cells [14,50,51].

Currently, increased attention to the surface markers of senescent cells is directed towards the search of therapeutic targets for their removal and reduction of inflammatory activity (Table 1).

For example, NOTCH1 and SCAMP4, involved in inflammation, were recently identified on senescent cells [67,69]. Some markers have already been used for the isolation of senescent populations, their further genetic analysis and as targets for various delivery platforms for senolytic therapy in animal experiments (Table 1).

A large number of senescent cells were found among memory cells and their number increases with age. Although the population of senescent immune cells is heterogeneous, memory cell markers can enrich sorted fraction with senescent cells. However, without functional tests, such as determination of the ability to proliferate upon antigenic stimulation, evaluation of cytokine production and cytotoxicity, and morphological features, it is difficult to identify senescent lymphocytes only by surface markers.

### 2.3. SASP

In 2000, Claudio Franceschi et al. proposed the concept of “inflammaging”—a state of chronic mild low-grade aseptic inflammation that develops as a result of constant antigenic load and stress during life, contributing to the weakening of immune cell activity and the pathogenesis of age-related diseases, including cancer [80,81,82]. Immune cells that have reached the senescence phase secrete various pro-inflammatory cytokines (TNF, IL-6, IL-1α/β, IFN-γ), chemokines (IL-8/CXCL8), MCP-1/CCL2, MIP-1α/CCL3, GROα/CXCL1), suppressive cytokines (TGF-β, IL-10), growth factors (GM-CSF, G-CSE, VEGF), matrix metalloproteinases (MMP-1, MMP-3, MMP-10), soluble receptors and ligands (ICAM-1/3, Fas, EGF-R), angiogenic factors, and other compounds that are now combined under the common term SASP (senescence-associated secretory phenotype) [83,84,85]. In addition to increased concentrations of TNF, IL-1β, and IL-6, an increase in plasma levels of C-reactive protein (CRP) is typical of the “inflammaging” state [86,87,88]. The accumulation of neopterin, secreted by macrophages in response to IFNγ stimulation, correlates with age regardless of gender and has been observed in physiologic aging and autoimmune diseases; therefore, it is also considered as another biomarker of age-associated chronic inflammation [89]. Recently, a pronounced correlation with the age of urinary neopterin concentration has been shown for rhesus macaques [90]. Thus, the senescent state is characterized by inadequate sterile inflammation with a simultaneous loss of the ability to activate in response to specific stimuli, such as pathogen invasion. The hypothesis of «trained innate memory» has been put forward as an explanation for the controversial production of proinflammatory factors by dysfunctional immune cells. According to the hypothesis, innate immune cells are capable of maintaining a stable state of activation in the absence of specific stimulation, which is associated with global metabolic and epigenetic changes [91]. Thus, SASP provides a heterogeneous environment combining pro-inflammatory and suppressive cytokines as well as DAMPs (danger-associated molecular patterns) and, therefore, has pleiotropic effects.

Selective inhibition of certain SASP components along with the use of senolytics is considered as one of the strategies to combat age-associated diseases and destructive effects of aging processes, including the immune system [92]. Some drugs targeting different components of SASP, such as therapeutic monoclonal antibodies against IL-6 (Tocilizumab) and TNF (Infliximab), are already used in clinical practice [93,94,95]. Various methods for determining the production of pro-inflammatory cytokines comprising SASP are widely used in the detection and characterization of senescent immune cells [96].

### 2.4. Decrease in Telomere Length and Telomerase Activity

The “Hayflick” cell division limit is the number of times a normal cell divides before entering the senescent phase (usually 40–60). With each division, a decrease in the length of telomeres occurs until short telomeres cause DNA instability and the induction of a program of accelerated cellular senescence. Telomere length is a recognized marker of aging that correlates with chronological age [97,98].

The molecular mechanisms of a senescent phenotype emergence in response to telomere shortening remain poorly understood. However, it is known that chromosome fusion due to telomere dysfunction initiates DDR signaling and leads to the accumulation of chromatin fragments in the cytoplasm. These cytosolic DNA fragments containing DSBs are recognised by cyclic GMP-AMP synthase (cGAS) and activate the Stimulator of Interferon Genes (STING), leading to premature aging [99]. Furthermore, the activated cGAS-STING pathway induces transcription of SASP components [100].

Age-dependent telomere shortening occurs in all types of lymphocytes, including T-, B-, and NK cells [101], and is associated with the risk of age-related and autoimmune diseases [102,103,104], susceptibility and severity of infectious diseases [105,106], as well as mortality [107,108]. Telomerase activity and telomere length are higher and the rate of telomere shortening is lower in the T-cells of long-lived individuals and correlate with the health status of centenarians [109]. However, the length of telomeres and the rate of telomeres shortening vary considerably among individuals, across different tissue types [110] and even within the same organ [111]. In addition, in healthy individuals, peripheral blood leukocytes are a highly heterogeneous population and their cellular composition is highly variable [112], which determines the highest variability in telomere length in each cell type in the lymphocyte population [111]. The length and rate of telomere shortening differ in T- and B-cells, and monocytes [113]. In the study by Lin et al., the highest telomerase activity and the longest telomere length were observed in B-lymphocytes. CD4+ T-lymphocytes had slightly higher telomerase activity than CD8+CD28+ T-lymphocytes; however, they had comparable telomere length [114]. The average telomere length for naïve CD4+ or CD8+ T-cells is ~2.5 kb longer than that of effector or memory T-cells [115]. However, during antigenic stimulation of T-lymphocytes, a transient increase in telomerase expression and telomere elongation is observed [116].

### 2.5. Cell Cycle Arrest and Expression of p16, p21 and p53

It is well known that the senescent state is characterized by cell cycle arrest. Inhibitors of cyclin-dependent kinases p16 and p21, as well as p53 protein, which are important elements of tumor suppressor pathways, are directly involved in its regulation and, consequently, in the control of cellular senescence [117,118]. Activation of the p53 pathway by phosphorylation has been shown to occur primarily in response to DNA damage (DNA damage response, DDR) and telomere dysfunction, whereas the p16 pathway is activated in response to mitogenic stress and reactive oxygen species (ROS) accumulation, exposure to oncogenes (e.g., RAS), and general cellular stress (including contact inhibition, cycling exhaustion and suboptimal culture conditions) [119]. In 2009, Liu Y. et al. showed that p16 expression is a reliable biomarker of human chronological age. The highest and most stable expression of p16 is observed in CD3+ T-lymphocyte from elderly donors that probably suggest accumulation of senescent cells in peripheral blood. Bad habits and lifestyle, such as smoking and reduced physical activity, dramatically increase p16 transcript expression levels. The study found that over the age of 60 (by the age of 80), the level of p16 expression increases, on average, almost 10-fold, while telomere length typically decreases by less than half over the same period of time [120].

Increased expression of p16 and p21 is now considered a recognized marker of senescence and is widely used in various studies, including immune cells [121]. It has been shown that physical activity decreases p16 and p21 expression in CD3+ T-lymphocytes [122]. An important feature of p16 and p21 as markers of senescence is that their increased expression is not detectable in the exhaustion state, another dysfunctional cellular state distinct from classical senescence described in the literature [9].

### 2.6. Metabolic Changes/Disorders

Age-related decline in immune system function and surveillance are also associated with changes in cellular metabolism. Metabolic disorders significantly reduce the amplitude of the adaptive immune response, reducing resistance to viral infections [123]. Senescent cells are characterized by inflammatory metabolism, secreting matrix proteases and pro-inflammatory factors. This type of metabolism reduces their sensitivity to antigenic stimuli and turns them off from the physiological immune response. The accumulation of senescent cells with age induces a change in the metabolism of the whole organism, both blocking the local immune response and slowing down regeneration in the surrounding tissues [124].

#### 2.6.1. Energy Metabolism Disruptions

The energy supply of quiescent (non-proliferating) naïve lymphocytes is provided by oxidative phosphorylation (OXPHOS) [125], whereas activation and differentiation of T-cells into effector cells is associated with the switch to the less energetically favorable process of glycolysis (aerobic glycolysis, known as the Warburg effect), which is probably necessary for optimal cytokine production [126]. The transition of effector T-cells to a quiescent and memory state is characterized by resumption of OXPHOS, apparently mediated by IL-15 [127]. In contrast to memory cells, senescent cells also carry out aerobic glycolysis through activation of the mTOR signaling pathway in response to DNA damage via induction of the two major transcription factors HIF1α and c-MYC. In addition, transcription of genes involved in the pentose phosphate pathway and de novo lipogenesis is enhanced [128,129]. It is interesting that EMRA effector CD8+ memory T-cells (CD45RA+CD27+/−), comprising the senescent population, show reduced proliferative potential, accumulate with age and are associated with several diseases, characterized by preferential energy acquisition through glycolysis in contrast to the EM (CD45RA+/−CD27−) population, which combines glycolysis and OXPHOS [130]. Expression of the suppression and exhaustion marker PD-1 on the surface of activated T-lymphocytes inhibits glycolysis and induces the switch to lipolysis, allowing endogenous free fatty acids to be utilized in the β-oxidation process during the reduced ability of cells to absorb and utilize other classes of nutrients [131]. Thus, the exhausted state of lymphocytes is characterized by a weakening of glycolysis, while the senescent state is characterized by its enhancement. At the same time, naive CD4+ T-cells from old mice show a significantly reduced rate of oxygen consumption when stimulated with anti-CD3/anti-CD28 antibodies and, therefore, exhibit impaired glycolysis, although impaired mitochondrial respiration is often compensated for by enhanced glycolysis. Senescent naïve T-cells reveal lower levels of basic intermediates of glycolysis, the pentose phosphate pathway, and the tricarboxylic acid cycle (TCA) [132]. Thus, the senescent state is simultaneously characterized by a complete switch from OXPHOS to aerobic glycolysis and a general slowing of metabolism, respiration, and protein biosynthesis. In normal lymphocytes, in response to TCR stimulation and CD28 co-stimulation, GLUT1 receptor controlling glucose uptake is upregulated to meet increased energy requirements during activation and effector functions [133,134]. A decrease in GLUT1 receptor expression on aged T-cells has been shown [130], which, in combination with loss of CD28 expression, leads to glucose deficiency and is likely another factor limiting the functionality of senescent lymphocytes. In the context of a significant impairment of glucose metabolism in senescent cells, it is not surprising that an increased population of CD8+CD28-CD57+ T-cells was found in diabetic patients. An increase in the CD4+CD28-CD57+ T-lymphocyte population was also observed in diabetic patients but did not reach statistical significance. CD8+CD28-CD57+ T-lymphocytes show signs of senescent metabolism, including elevated concentrations of reactive oxygen species (ROS) and ECAR (extracellular acidification rate) due to increased glycolysis [15].

#### 2.6.2. Mitochondrial Dysfunction

Decreased Ca^2+^ uptake by mitochondria of T-lymphocytes during aging impairs Ca^2+^ -dependent signaling and downstream induction of crucial pro-inflammatory transcription factors, such as NFAT1 and NF-κB [135]. Naive CD4+ T-lymphocytes from old mice show reduced Ca^2+^ ion uptake in response to stimulation by TCRs, consequently reducing T-cell proliferation and IL-2 production capacity [136]. Memory T-cells derived from such old naive CD4+ T-lymphocytes are also dysfunctional, showing reduced IL-2 production and impaired response upon re-stimulation [137]. However, senescence can also be caused by the excessive mitochondrial activity. For example, the accumulation of sphingolipid ceramides CerS6/C14 in the outer membrane due to increased expression of ceramide synthase over-activates mitophagy in T-cells, causing increased mitochondrial fission and senescence-like cellular dysfunction. The inhibition of ceramide metabolism prevents excessive mitophagy and restores the central memory phenotype in these cells [138]. Reduced mitochondrial function may cause senescence due to p53 overexpression resulting from telomere dysfunction through inhibition of Pgc1 (peroxisome proliferator-activated receptor gamma coactivator 1-alpha) [139].

Mitochondrial dysfunction is a hallmark of lymphocyte senescence. A decrease in the respiratory capacity and membrane potential of mitochondria, accompanied by the production of ROS, is the reason and also a consequence of aging of adaptive immune system. Age-dependent changes in mitochondria can be proposed as targets for the development of new, potent senolytics for amelioration immune function.

#### 2.6.3. Autophagy and Mitophagy Disorder

A stable level of autophagy clears the cell of damaged membranes and misfolded proteins and keeps cells healthy. A decrease in the level of autophagy is characteristic of senescent cells. With age, cells also accumulate damaged and poorly functioning mitochondria. However, the process of mitophagy (digestion of damaged organelles) can both remove damaged mitochondria and stimulate mitochondrial renewal through division due to lack of energy supply. Healthy mitochondria neutralize reactive oxygen species and supply cells with enough high-energy molecules to power the cell’s biochemical machinery. Selective damage to mitochondria or inhibition of autophagy accelerate cellular senescence [140,141].

Tai H. et al. showed that autophagy and mitophagy are related processes. Mitochondrial dysfunction precedes lysosomal dysfunction, which in turn leads to impaired autophagy. As a result, the number of damaged membranes, organelles and other intracellular structures and lipofuscin increases in senescent cells [142]. Although the number of mitochondria remains unchanged in CD4+ T-lymphocytes from donors of different age groups (in both naive and memory cells), a significantly larger number of autophagosomes, containing undegraded mitochondria, is detected in elderly individuals [143].

The disruption of autophagy correlates with increased intracellular ROS levels and inflammatory factor upregulation. The blockade of autophagy by Atg5 knockdown increases SA-β-Gal positive cell number, elevates IL-6 secretion, and induces senescence [142]. Thus, disruption of autophagy itself causes cellular senescence. At the molecular level, the most important inhibitor of autophagy is mTOR [144,145]. Moreover, mTOR-dependent autophagy activators (rapamycin and PP242) dramatically reduce the percentage of SA-b-Gal—positive cells, which correlates with decreased IL-6 levels, while the mTOR-independent autophagy activators valproic acid (VPA) and LiCl have only minor effects on senescent cells [142]. In addition, mTOR inhibition reduces p16, p21, and p53 levels [145,146] and decreases SASP production [145,146,147]. Therefore, not only the enhancement of autophagy, but also the inhibition of other effects of mTOR are important in preventing the cell senescence program.

Thus, the senescent state is characterized by large-scale metabolic, mitochondria, autophagy, and mitophagy dysfunction. Senescent lymphocytes contribute to chronic inflammation and reduced immune system function in the elderly. Given the central role of mTOR in these processes and the positive effects of its inhibitors on senescence markers, targeting this signaling pathway should be considered as a promising approach to develop anti-aging and immunomodulatory therapeutic approaches for healthy longevity.

#### 2.6.4. SA-β-Gal

In healthy mammalian cells, pH 4.0 is optimal for lysosomal β-galactosidase activity. In 1995, Dimri et al. demonstrated that during replicative senescence of keratinocytes and fibroblasts, the optimum activity of this enzyme shifts to pH 6.0. In addition, this form of the enzyme is absent in pre-senescent and quiescent fibroblasts, as well as in terminally differentiated keratinocytes and proliferating cells. This β-galactosidase, which is active at pH 6.0, has been termed senescence-associated β-galactosidase (SA-β-Gal) [148]. It was first hypothesized that SA-β-Gal might be an alternatively spliced form of lysosomal β-galactosidase, whose presence is attributed to the increased unspecific lysosomal activity found in senescent cells with aging [148].

However, the view later developed that β-galactosidase activity at pH 6.0 is associated with the accumulation of specific glycoproteins and complex glycolipids in senescent cells [149], such as lipofuscin granules (see above) and other lipoprotein complexes, and derived by disruption of cellular metabolism. It was shown that SA-β-Gal activity as a marker of senescence effectively reflects the completeness of cell immortalization and can be used to monitor their state at different stages of immortalization of cell lines. For instance, in cells derived from human ovarian surface epithelial cells (HOSE 6-3), a dramatic decrease in SA-β-Gal activity was observed after these cells acquired immortalized status. In addition, an inverse relationship between telomerase activity and SA-β-Gal was shown in immortalized cells [150]. Martínez-Zamudio R. I. et al. showed that the activity of SA-β-Gal is a specific marker of senescence including immune cells, suitable for efficient identification of senescent lymphocytes of different populations of human PBMCs. The highest correlation between SA-β-Gal expression and age was observed for the CD8+ T-cell population. In donors aged 57–67 years, the proportion of SA-β-Gal-positive CD8+ T-cell reached 64 ± 4% (Mean ± SEM) and correlated with age-related immune system disorders [8]. The histochemical detection of SA-β-Gal activity can be used in various studies such as evaluating the influence of tumor microenvironment tissue aging and the induction of immune cell senescence.

Jian Ye et al. used the detection of SA-β-Gal activity, along with other senescence biomarkers, to investigate the mechanisms of immunosuppressive effects of breast tumor-derived γδTregs. When these γδTregs were co-cultured with fractions of PBMCs, they induced a senescent state in naïve and effector T-lymphocytes, as well as dendritic cells [151]. By chromogenic SA-β-Gal staining after co-cultivation of the periodontal pathogen *P. gingivalis* with bone marrow-derived mouse dendritic cells, Elsayed R. et al. found an accelerated cellular senescence [152].

Despite the efficiency of SA-β-Gal as a senescent marker for most cell types, it is important to note that increased lysosomal beta-galactosidase activity is characteristic of some cells, such as active macrophages, Kupffer cells, and osteoclasts, in the normal state [153,154]. Moreover, a significant number of NK cells demonstrated a high level of SA-β-Gal signal regardless of donor age. This high signal level may indicate an increase in lysosome-associated secretory organelles rather than a sign of senescence [8]. Thus, given these limitations and in combination with other documented characteristics of senescence, SA-β-Gal may serve as a useful biomarker of senescence in diverse peripheral blood lymphocyte populations.

### 2.7. Disorganization and Dysfunction of Chromatin

#### 2.7.1. HMGB1

Extremely conserved, high-mobility group protein B1 (HMGB1) is a non-histone protein with 99% identity among mammals. HMGB1 has two homologous DNA-binding domains and, when localized in the nucleus, binds to the small groove of B-type DNA, albeit with limited specificity, forming a 90° or more bend in the DNA double helix [155,156]. As a so-called chromatin architectural factor, HMGB1 directly interacts with a number of proteins such as transcription factors containing HOX or POU domains, p53, NF-kB and steroid hormone receptors, promoting their recruitment and facilitating interactions between these proteins and DNA. Furthermore, HMGB1, through its interaction with proteins that activate the RAG1/2 gene, is involved in enabling the V(D)J recombination process by enhancing specific recognition and facilitating DNA cleavage [157,158]. This may be of particular interest in the context of the well-known fact that the diversity of the T-cell receptor repertoire (TCRs) decreases with age [159,160]. Taking into account the loss of nuclear localization of HMGB1 during senescence and the important role of this protein in the process of V(D)J recombination, it can be assumed that this phenomenon may be one of the factors and mechanisms leading to a decrease in the repertoire of TCRs of T-lymphocytes maturing in the thymus with age.

Another important function of HMGB1 is intercellular signaling and its role as an alarmin. It is known that passive release of HMGB1 occurs during necrotic cell death, whereas in apoptosis HMGB1 remains bound to chromatin until it is eliminated by macrophages or non-professional phagocytic cells—scavengers [161]. Activated macrophages, monocytes, and dendritic cells also serve as a source of extracellular HMGB1. HMGB1 danger signaling to surrounding cells occurs through interaction with its receptors RAGE, TLRs 2, 4 and 9, syndecan and thrombomodulin, followed by activation of the NF-κB signaling pathway [158,162,163]. It has been shown that secretion of HMGB1 into the extracellular medium is accomplished by its acetylation on lysine residues. Under the influence of lipopolysaccharide on monocytes and macrophages, or under the exposure to trichostatin A histone deacetylase inhibitor (HDAC) on quiescent macrophages, HMGB1 is hypersacitylated and translocated to the cytosol with subsequent accumulation in secretory lysosomes [164]. Extracellular HMGB1, as a ligand of TLRs, has been shown to promote sterile inflammation through the induction of IL-6, a key component of SASP. An important role of this protein in the production of inflammatory cytokines by immune cells has been described. The transition to cellular senescence due to telomere shortening, genomic instability, or DNA damage leads to loss of nuclear localisation of HMGB1 and increases the protein level in the extracellular space [165], which enhances TLR/NF-κB-dependent production of SASP components [166].

Senescence induced by X-ray irradiation, replicative depletion, or overexpression of p16 and the RAS oncogene results in a significant decrease in HMGB1 nuclear localization and its migration to the cytoplasm. Consequently, loss of nuclear HMGB1 characterizes the senescent state of cells regardless of the senescence inducer, but p16 overexpression by itself is not an activator of HMGB1 migration to the cytoplasm. Both depletion and overexpression of HMGB1 stimulated p53 expression in human mammary epithelial cells and mouse embryonic fibroblasts to the levels found in cells subjected to irradiation-induced or replicative senescence. Blocking p53 by RNA interference ensured the preservation of HMGB1 nuclear localization during X-ray irradiation-induced fibroblast senescence. Thus, p53 activity is regulated, among other things, by the expression level of HMGB1, and, at the same time, HMGB1 re-localization is directly dependent on p53 activity (in contrast to the expression of SASP components). In addition, decreased nuclear and serum HMGB1 was observed in vivo in old but not young mice [166].

Lee J. J. et al. showed that highly metastatic B16-F10 cells (mouse melanoma) mainly showed signs of cellular senescence and exhibited HMGB1 expression in response to genotoxic stress (doxorubicin treatment); in contrast, poorly metastatic cells entered apoptosis, exhibiting decreased HMGB1 expression levels. Moreover, depletion of HMGB1 in B16-F10 cells caused transition from senescence to apoptosis with decreased p21 expression, while HMGB1 overexpression led to transition from apoptosis to senescence with a corresponding increase in p21 expression after induction of genotoxic stress due to doxorubicin exposure [167]. This suggests a key role for HMGB1 in the choice of cellular response strategy to stressors and the emergence of a senescent state.

Thus, HMGB1 is a pivotal regulator of cellular senescence, participating in the direct maintenance of normal chromatin function and indirect stimulation of SASP as a danger signal, which also plays a role as a regulator of signaling cascades in response to various stressors. In this regard, it is extremely interesting to validate this marker for identifying senescent immune cells.

#### 2.7.2. SAHF

In 2003, Narita et al. showed that senescent human fibroblasts are characterized by the formation of foci of a previously undescribed form of facultative chromatin enriched in heterochromatin protein 1 (HP1) and heterochromatin histone H3 trimethylatation at lysine 9 (H3K9me3), which provides a binding site for HP1. The euchromatic markers H3K9 and H3K4me3 are absent in these foci. These senescent specific foci are collectively called SAHF—senescence-associated heterochromatic foci. The accumulation of SAHF-positive cells after oncogene-induced senescence by RAS overexpression correlates well with the kinetics of other senescence markers such as senescence-associated beta-galactosidase (SA-β-gal) activity, p16 expression, Rb hypophosphorylation, and cell cycle arrest. RAS-induced senescence, SAHF formation, and SA-β-gal activity depend significantly on the activation of the p16/Rb pathway, while the influence of p53 on this process is minor. The knockdown of p16 or Rb substantially suppresses SAHF formation in RAS-induced senescence, but these p16 or Rb-deficient cells accumulate and have senescent morphological features and SA-β-gal activity [168]. However, SHAF formation is not a universal marker of cellular senescence, characterizing the senescent state independently of cell type and stress exposure. Kosar M. et al. showed that primary human fibroblasts (BJ and MRC-5) and primary keratinocytes enter a senescent state under the influence of replicative stress when the Ras oncogene is overexpressed, or as a result of DNA or cellular damage caused by radiation, chemotherapy (e.g., doxorubicin, etoposide, hydroxyurea) senescence caused by telomere attrition, or bacterial cytolethal distending toxin. All primary cell lines tested formed SAHF in response to replication stress. However, when senescence is induced by genotoxic stress or cellular stress agents, only MRC-5 forms SAHF, but not the other primary lineages tested [169]. In addition, SAHF formation does not occur in all senescent cells. For example, in senescent human breast cancer MCF7 and fibrosarcoma U2OS cells, SAHF is invisible upon induction by γ-irradiation or inhibition of the replication and Chk1 or ATR kinases [170,171]. SHAF accumulation by senescent lymphocytes and other immune cells has not been studied, but such data could be useful to better understand the mechanisms of immunosenescence.

#### 2.7.3. Lamin B1

The anchoring of heterochromatin on the nuclear lamina is an important element in ensuring the spatial organization of chromatin structure and the functioning of eukaryotic genomes. Inner nuclear membrane proteins are able to recognize specific protein lamina-associated domains (LADs) and bind lamins A/C or B, respectively. For lamin B1, such a protein is its receptor LBR [171].

Lamin B1 (LB1) expression is known to be decreased during replicative and oncogene-induced senescence in various cell types [171,172,173]. The induction of senescence also results in decreased LBR expression [174]. The decreased expression of LB1 does not occur directly in response to DNA damage (DDR), activation of mitogen-activated protein kinase p38 (p38-MAPK) and nuclear factor-κB (NF-κB) or reactive oxygen species (ROS), which are all hallmarks of many senescent cells, but is observed upon direct stimulation of the p53 or p16/pRB signaling pathways: an elevated level of p53 or p16 expression was found to be a sufficient condition for the loss of LB1 [172].

Sadaie M. et al. showed that lamin B1 is depleted predominantly from the central regions of lamina-associated domains enriched with the repressed chromatin H3K9me3 tag during RAS-induced senescence. The knockdown of lamin B1 facilitates spatial re-localization of perinuclear H3K9me3 positive heterochromatin, which may be prevented by ectopic expression of LB1. At the same time, increased LB1 binding was observed in small regions of the genome enriched in H3K27me3 and associated, respectively, with gene repression. LADs were shown to be enriched in H3K27me3 at the edges, whereas H3K9me3 occupies the entire LAD. In addition, the loss of LB1 is associated with the formation of SAHF. The depletion of H3K9me3-rich regions by LB1 and its spatial redistribution presumably contributes to the creation of a “pro-SAHF” nuclear environment [175].

Thus, the involvement of LB1 in the development of cellular senescence may be related to at least two factors: uneven redistribution throughout the genome and spatial reorganization of chromatin, as well as gene repression [175]. The loss of LB1 caused by disruption of the spatial structure of chromatin containing LADs could be responsible for cell senescence and loss of cell proliferation capacity through p53- and Rb-dependent mechanisms [171].

Regarding immune cells, it has been shown that exposure to tau protein induces loss of the nuclear envelope protein LB1 and the histone marker of H3K9me3 in microglia cells, which may indicate a role of LB1 and HMGB1 in the development of neurodegenerative diseases with age [176,177].

#### 2.7.4. γH2AX

Apart from the canonical histone H2A, the histone variant H2AX is widely represented in mammalian cells, accounting for about 2.5–25% of total H2A [178]. The appearance of its phosphorylated form, called γH2AX, is one of the earliest events in response to DNA damage (DDR), such as various genotoxic stresses that induce double-strand breaks (DSBs). The occurrence of DSBs activates ATM, ATR, and DNA-PK kinases of the PI3K (phosphotidylinositol-3-kinase) family, which carry out phosphorylation of H2AX at the serine residue at position 139 (ser139). γH2AX foci can be detected in cell nuclei as early as 3 min after irradiation; then their number reaches a maximum within 30 min and remains unchanged for up to 60 min. The total number of γH2AX foci correlates with the total number of DSBs, and the size of γH2AX foci in nuclei at 3 min after irradiation is smaller than at 15 min. During DSBs repair, several phosphatases, such as PP2A, PP4, Wip1, and PP6, carry out the dephosphorylation of γH2AX [179].

The formation and accumulation of γH2AX foci, which indicate the accumulation of persistent lesions and unrepairable double-stranded DNA breaks, respectively, were found to occur with increasing passage in various mouse and human cell lines, as well as resulting from chemical exposure (hydrogen peroxide). It has also been shown that γH2AX foci co-localize with repair factor proteins such as 53bp1, Mre11, Rad50 and Nbs1 [180]. Rodier F. et al. showed that the formation of γH2AX foci as a result of high-dose radiation exposure is able to initiate the secretion of IL-6 and IL-8, essential components of SASP, in human HCA2 and WI-38 fibroblast cultures. Their secretion increased 5–6-fold within 2–4 days and reached levels seen in replicative senescence within 3–5 days. The authors concluded that DNA damage itself did not lead to the development of inflammatory response and cytokine secretion; it happened with some delay when sufficient damage accumulated for the stable activation of downstream molecules of the DDR signaling cascade to occur. This assumption is supported by the fact that cells with a large amount of DNA damage foci show high levels of IL-6 and IL-8 secretion, while cells with a small amount of damage are characterized by low levels of their production during senescence induced by p16 overexpression. Nearly complete depletion of ATM kinase abolishes the increased IL-6 expression observed 9–10 days after X-ray irradiation and, moreover, cancels the increased IL-6 production in cells already undergoing replicative senescence [181]. Depletion of H2AX itself has a similar effect, suppressing IL-6 secretion both 2–3 days after induction of senescence by irradiation and 9–10 days later, when inflammatory cytokine secretion becomes more pronounced [182]. It has been shown that CD8+ T-lymphocytes containing γH2AX do not respond to stimulation with IFN-α, IL-2, or IL-6 [183].

Along with other hallmarks of cellular senescence, γH2AX is widely used as a marker of senescent cells, including immune cells. Among the obvious advantages is the fact that γH2AX foci are formed within seconds after DSB formation; however, since they are initially rather small and difficult to visualize, for more reliable detection they are examined after 15–30 min in the case of senescence induced by irradiation. Another advantage of γH2AX as a senescence marker is that its distribution covers a fairly extensive chromatin region of several megabases in size either side of the DSB site, so they can be easily visualized with specific antibodies, with virtually every γH2AX foci corresponding to a single double-stranded break. Furthermore, an important feature of γH2AX is that as soon as DSB repair starts, dephosphorylation of γH2AX molecules is also triggered, leading to the elimination of its foci. The number of visible γH2AX foci allows to track DNA repair processes over time and monitor the accumulation of unrepairable double-strand breaks. A major limitation of detecting DSBs by the presence of γH2AX is the formation of non-double-stranded γH2AX foci on ssDNA sites generated during DNA replication [184]. Obtaining clear data on the expression of these markers on senescent lymphocytes will improve our understanding of the causes of aging of the immune system and methods of correction.

### 2.8. Multi-Omics Changes

#### 2.8.1. Transcriptome

The list of hallmarks of senescent cells would not be complete without a description of the transcriptional profile, which has been actively studied using single-cell RNAseq technologies that have recently become widespread. Casella G. et al. compared the transcriptomes of several types of human fibroblasts (WI-38, IMR-90, HAEC and HUVEC) in different types of senescence (replicative or induced by irradiation, doxorubicin and oncogenes) and found increased expression levels of 50 and decreased levels of 18 total transcripts. The most repressed gene was MCUB (Mitochondrial Calcium Uniporter Dominant Negative Subunit Beta) [185], consistent with mitochondrial dysfunction and impaired regulation of mitochondrial calcium uptake during senescence, particularly as observed by Ron-Harel N. et al. in CD4+ T-lymphocytes [186]. It has been established that suppression of the expression of nuclear fibrillarin methyltransferase (FBN) rRNA disrupts ribosome assembly and protein biosynthesis, which may be one of the reasons for the inability of senescent cells to perform their functions [185]. FBN is an oncogene and its increased expression is characteristic of different types of tumors, which contributes to a significant increase in the rate of tumor proliferation. In addition, activation of the tumor suppressor p53 directly suppresses FBN expression [187]. Thus, FBN suppression may be a key mechanism for cell cycle arrest in senescent cells. Upon induction of senescence, fibroblasts also exhibited reduced expression levels of prothymosin-α (PTMα) [185], an important regulator of cell proliferation and apoptosis and, in addition, a documented biomarker of tumor aggressiveness in various malignancies [188]. Accordingly, reduced expression of PTMα is likely another reason for the loss of senescent cells’ ability to proliferate and resistance to apoptosis. A further characteristic feature of cellular senescence, confirmed by transcriptome analysis data, is the decreased expression of genes encoding histones, such as HIST1H1D, HIST1H1A, HIST1H1E and HIST2H2AB [185].

Among the transcripts whose levels increased upon induction of senescence in a different type of fibroblasts, Casella G. et al. found SASP component TNF and its receptor (senescence is known to be associated with an inflammatory background) and the Rho family GTPase 3 (explaining the characteristic changes in the morphology of senescent cells) [185]. Increased levels of Nicastrin, which acts as an oncogene through activation of the NOTCH and PI3K/Akt signaling pathways and inhibition of apoptosis, have also been found [185,189,190].

Increased expression of these genes may reflect the cell’s attempt to escape from a state of cell cycle arrest by activating compensatory mechanisms. In addition, the activation of protoncogenes detected in cellular senescence can potentially induce mechanisms of oncogenesis, which, together with large-scale disorganization of chromatin and dysregulation of gene expression, confirms the connection between aging processes and malignant transformation of cells. Induction of fibroblast senescence increases the expression level of PURPL (lncRNA), which suppresses p53. PURPL is more actively produced in response to increased p53 levels. Remarkably, PURPL transcript levels were higher than p16 and p21 mRNA levels, so it could serve as a new, more robust transcript biomarker correlated with age and cellular senescence [185]. Distinctive transcriptomic profiles of senescent cells have been described for different types of fibroblasts, but scRNAseq data for senescent immune cells are lacking, despite a large number of bulk RNA-seq studies of PBMC from donors of different age groups [191,192,193].

MicroRNAs may be among the key regulators of T-cell differentiation, development and activation. For example, one study showed a decrease in miR-181a expression in relation to age [194]. miR-181a is one of the most abundant microRNAs in lymphocytes, its loss is associated with impaired T-cell differentiation and may serve as an additional marker of senescence.

Thus, studying the transcriptomes of senescent cells in various populations of human peripheral blood will allow a better description of the aging process of the immune system and may become one of the central directions of further research in this area.

#### 2.8.2. Epigenetic Changes

The aging process is accompanied by large-scale demethylation of genomic DNA [195]. Recently, a large amount of data has been accumulated on specific CpG islands, the hyper- or hypomethylation of which significantly correlates with chronological age [195,196,197,198,199,200]. However, changes in methylation levels often do not correspond to differential gene expression. For example, Steegenga W. T. et al. observed age-dependent methylation of ELOVL2, FHL2, PENK and KLF14, previously reported epigenetic biomarkers of senescence, but these genes themselves did not show age-related changes in expression. Accordingly, DNA methylation does not always determine differential gene expression during aging. However, a correlation was found between increased TNF expression and loss of promoter methylation in PBMC among immune response-related genes [201], consistent with its central role as a component of SASP.

Note that age-dependent changes in gene expression and DNA methylation landscape also affect the earliest populations of blood cell progenitors, including hematopoietic stem cells (HSCs), which often biases hematopoiesis toward preferential differentiation in one direction (e.g., myeloid). In addition, mutations cause epigenetic changes inducing clonality, cellular stress, displacement of the normal development of the hematopoietic system and the accumulation of senescent cells [202,203].

#### 2.8.3. Chromatin Accessibility

Developed about a decade ago, the latest technology for full genomic assessment of chromatin accessibility ATAC-seq (Assay for Transposase-Accessible Chromatin using sequencing) allowed additional characterization of chromatin loci of immune cells that are differentially open or closed depending on age [204]. Chromatin closing with age largely affects T-lymphocyte activation genes (161 genes) and TCR signaling pathways (59 genes). CD4+ T-lymphocytes (both naive and memory cells) were found to exhibit significantly less chromatin remodeling during senescence compared to CD8+ T-cells. Moreover, memory CD8+ T-cells demonstrated stronger and more extensive chromatin remodeling compared to naïve CD8+ cells. Memory CD8+ T-cells undergo global promoter repression with age, whereas naïve CD8+ T-cells show a loss of accessibility mainly of enhancers. In general, the population of CD8+ memory T-cells exhibits the most significant chromatin remodeling with age.

Chromatin closure and associated reduction in expression of the homeostatic cytokine receptor IL-7 (IL-7R) gene were found in CD8+ T-lymphocytes, affecting both the naïve population and central and effector memory cells. This results in a decreased sensitivity of CD8+ CTLs to IL-7 with aging, which is not observed in the CD4+ lymphocyte population. Genes downstream of IL-7R signaling cascade, such as JAK1, JAK3, STAT5A, STAT5B, and PTK2B, also exhibit chromatin closure with age. Chromatin closure is also specific for reduction of signal transduction through TCRs and for IL-2, IL-9 signaling [191,204]. Importantly, chromatin closure was characterized by downregulation of histone genes (e.g., HIST1H3D, HIST1H3E, HIST4H4) as well as histone modifier genes (e.g., EZH1, SETD7) [204]. Importantly, chromatin closure was characterized by decreased expression of histone genes (e.g., HIST1H3D, HIST1H3E, HIST4H4) as well as histone modifier genes (e.g., EZH1, SETD7) [204]. Chromatin closure is specific to senescent fibroblasts, which is also accompanied by a decrease in the expression of core histones and disruption of histone modification patterns [185]. While chromatin accessibility profiles have been demonstrated for distinct populations of senescent immune cells, transcriptome analysis and epigenetic changes similar to those observed in fibroblasts have not yet been studied for lymphocytes at the single cell level.

## 3. Conclusions

The accumulation of senescent cells is observed during the aging process. Senescent immune cells undergo multiple abnormalities, which affect the overall function of the immune system, causing inflammation and immune balance disruption.

The accumulation of DNA damage and unrepaired DSBs (as indicated by γH2AX staining), telomere shortening, oncogene activation and oxidative stress activate DDR signaling in the cell, which leads to cellular senescence, cycle arrest and increased resistance to apoptosis. In addition to genomic instability, the epigenetic modifications of senescent immune cells lead to global dysregulation of gene expression and chromatin spatial structure disruption, including loss of HMGB1 nuclear localization and SAHF formation. Senescence is also accompanied by the widespread metabolic disorders. In response to stress, the mTOR signaling pathway becomes dominant and the transcription factors HIF1a and c-MYC are activated, which ultimately induces aerobic glycolysis. Global metabolic dysregulation impairs lysosomal and mitochondrial function and mito- and autophagy, resulting in the accumulation of lipofuscin granules. Cessation of intercellular interactions, morphological changes, cell cycle arrest, production of inflammatory cytokines and other SASP components are now recognized as common hallmarks of cellular senescence (Figure 1).

The hallmarks of senescence are to the same extent characteristic of tumor cells and largely coincide with the hallmarks of cancer. However, at present, most of the generally accepted biomarkers of senescent cells have been characterized and successfully used for fibroblasts and various tumor cell lines, while little information on their application for the specific detection of senescent immune cells is available.

The combination of several biomarkers reflecting different pathways of senescence induction may help to create a universal multi-biomarker of senescent immune cells for efficient isolation from mixed populations and development of immunization strategies.

According to comprehensive studies using various markers and methods, the population of cytotoxic CD8+ T-lymphocytes is the most susceptible to age-related changes, while CD4+ lymphocytes are less dramatically affected by senescence. Discovering the reasons for this phenomenon will undoubtedly help advance our understanding of the mechanisms of aging of the immune system and the role of its individual elements in the development of various pathologies while also explaining healthy longevity. Further study of the complex mechanisms that lead to weakening of the immune system functions with age will facilitate the development of novel approaches for treatment and prevention of age-associated diseases, laboratory diagnostics and special vaccines aimed at overcoming immune dysfunction in the elderly, which will improve the quality of life and help to ensure an active and healthy longevity.

## Figures and Tables

**Figure 1 ijms-24-15653-f001:**
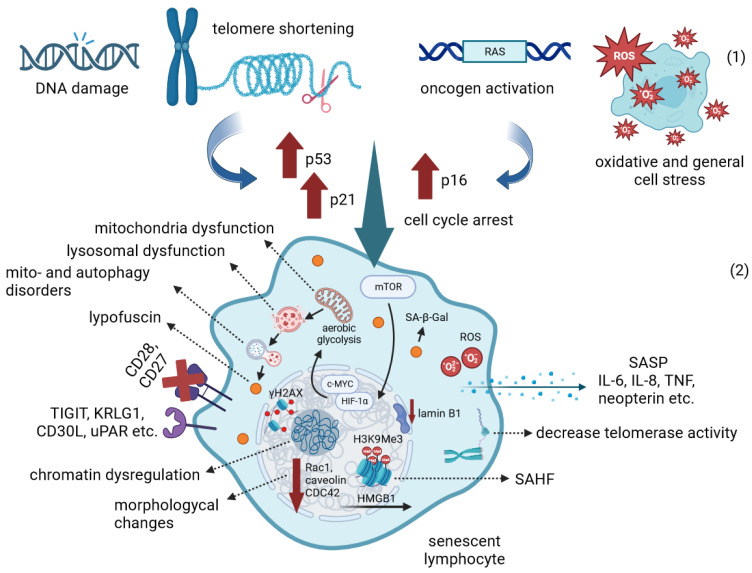
Cellular senescence: The first part (**1**) illustrates the triggers of the senescent state. The second part (**2**) shows multiple disregulations that cells undergo during senescence. The brown upward and down arrows show increased and decreased gene transcription, respectively. Dashed arrows show key characteristics of the senescent phenotype. Receptors recognized as surface markers of the senescent state are shown on the cell surface. The loss of surface CD27 and CD28 receptors observed in senescence is shown with a red cross. For more details, see the description in the text. Created with BioRender.com (accessed on 5 October 2023).

**Table 1 ijms-24-15653-t001:** Surface molecules expression biomarker variation upon senescence.

Surface Protein	General Function	Implication in Senescence/Aging	Potential Implication in Senotherapy	Refs.
CD28	Co-stimulatory molecules.	The loss of CD28 and CD27 expression on T-cells is the most consistent biological indicator of senescence in the human immune system, and the frequency of CD28- and CD27- T-cells is a key predictor of immune incompetence in the elderly.	Non-senescent cell exclusion marker	[38,39,41]
CD27			Non-senescent cell exclusion marker	[40,42]
CD57	A surface antigen that mainly characterizes T- and NK cells.	The CD57 is used to identify terminally differentiated “senescent” cells with reduced proliferative potential	Memory T- and NK cells, potential target for senolytics.	[44]
KLRG1	lymphocyte inhibitory co-receptor expressed predominantly on late-differentiated effector and effector memory CD8+ T- and NK cells	Biomarker of senescence. Decreases T-cell and NK function.	Potential target for senolytics. KLRG1 blockade reinvigorates T- and NK cells by correcting the impaired Akt (Ser473) phosphorylation.	[45,46,52,53]
TIGIT	Inhibitory co-receptor presented on some T- and NK cells	Biomarker of senescence. Decreases T-cell and NK function.	Potential target for senolytics. TIGIT blockade enhances T- and NK cell function.	[47,54,55]
CD148/DEP1/PTPRJ	Negative regulation of growth factor signalling and cell proliferation.	Biomarkers of senescence.	Potential target for senolytics.	[56]
B2MG/B2M	Presentation of peptide antigens to the immune system.	Biomarkers of senescence. High levels in serum of elderly.	Target for cytotoxic nanoparticles directed at senescent cells.	[56,57,58]
CD264/TNFRSF10D/TRAILR4	Antiapoptotic receptor, decoy receptor for TRAIL.	Markers of senescent hBM-MSCs.	Potential target for senolytics.	[59]
CD36	Scavenger receptor with a role in inflammation and lipid metabolism.	Regulation of lipid metabolism.	Required for initiation of SASP.	[60,61]
ICAM-1	Glycoprotein that mediates the adhesion between endothelial cells and activated leukocytes.	Marker of senescence. Increased expression in atherosclerotic lesions.	Oxidative stress-dependent increase. Potential target for senolytics in anticancer therapy. However, ICAM-1 is involved in the physiological endothelial inflammatory response.	[62]
MDA-Vimentin	Oxidized form of vimentin, an intermediate filament.	Marker of senescence. Increased expression in plasma of age-accelerated mice.	Oxidized form of vimentin, an intermediate filament is reliable senescence indicator.	[63]
CD26 (DPP4)	Cleavage of several substrates including cytokines and growth factors. Regulation of incretins in glucose homeostasis.	Biomarker of senescence. Protective role on the vascular system and kidney of aging mice.	Target for ADCC (NK-mediated cytotoxicity) for the clearance of senescent cells.	[64,65,66]
NOTCH1	Member of the NOTCH signaling pathway.	Regulation of different SASP profiles.	Regulation of SASP by small molecule application inhibition (γ-secretase inhibitor, PF-03084014).	[67]
NOTCH3	Member of the NOTCH signaling pathway.	Regulation of the onset of cellular senescence. Notch3 also regulates senescent cell survival. Notch3 signaling inhibits cell proliferation through upregulation PTEN.	Blocking NOTCH signaling with small molecules (e.g., γ-secretase inhibitor) reduces senescent cell survival and promotes clearance of senescent cells. However, immunosuppression, gastrointestinal bleeding, skin lesions and other side effects have been reported.	[68]
SCAMP4	Secretory protein involved in membrane trafficking.	Regulation of pro-inflammatory SASP.	SASP regulation.	[69]
NKG2D	Recognises proteins from the MIC and RAET1/ULBP families on the surface of stressed, malignantly transformed and infected cells	Increased expression in senescent and stressed cells	Potential target for senolytics. NKG2D has been used to eliminate senescent cells using CAR-T therapy. However, side effects such as hypophosphatemia, weight loss, hands and feet skin reactions, hypertension, etc., have been reported.	[70,71,72]
ULBP2 (MICA/B)	Ligands for the NKG2D receptor.	Regulation of immune surveillance.	Clearance of senescent cells through NK-mediated cytotoxicity. Side effects are similar to NKG2D.	[73,74,75]
uPAR	Regulation of intracellular signaling in response to extracellular components.	Upregulated in senescence.	Inflammatory response. It has been recently used to kill senescent cells using CAR-T therapy T-cells. Side effects of therapy may include systemic lesions and nephrotoxicity.	[76,77]
CD30L (CD153)	Pro-inflammatory cytokine of TNF superfamily, ligand of TNFRSF8/CD30 receptor. Expressed on activated immune cells.	Increased expression.	Vaccination against CD30L with a monoclonal antibody is used to block GvHD (graft-versus-host disease) activated Th CD4+ cells. It has recently been tested to remove aged immune cells with an inflammatory phenotype.	[78,79]

## Data Availability

Data available on request from the authors.

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
