# Peer review of "Reliable Hallmarks and Biomarkers of Senescent Lymphocytes"

_ijms, 2023, doi:10.3390/ijms242115653_

Round 1

Reviewer 1 Report

The work of Martyshkina et al. summarizes very throughly the hallmarks and biomarkers of senescent lymphocytes at different levels. They described morphological and molecular changes and expression of characteristic cell surface markers as well as alterations in cell signaling, metabolism, transcription and epigenetics. Moreover, the author summarized senescence-associated surface markers, which may represent a target for senolytics. This review is well-structured and in most parts clearly discussed.

Although the manuscript is of high quality, the Reviewer asks the authors to address the points listed below.

1)      Language: Readibility should be increased

-          Please mind throughout whole manuscript consistency in writing (esp. For the words aging (e.g. line 187), SA-β-Gal (e.g. line 319), IL-1β (e.g. line 162), NK cells (esp. Table 1), Ca2+ (line 283)

-          Mind correct use of articles (esp. „the“), singular/plural and punctuation.

-          Please ask a native speaker to proofread, with focus on word order. 

2) General remarks:

-          The Abstract starts with the statement: „Recently, the phenomenon of accumulation of senescent cells of adaptive immunity in the elderly attracting attention due to the increasing risk of global epidemics and the aging of the global population“. This fact is not very new. Please remove „Recently“.

-          Line 19:  CD8 T lymphocytes compared to CD4 à CD8+, CD4+

-          Line 74: The Reviewer doesn´t understand the sense of the word „including“ here

-          Line 80: Incomplete sentence or „observed in tissues“ is not necessary here

-          Line 250: EM and EMRA should be listed in Abbreviations

-          Line 504: LB1 HMGB1 à What do the authors mean?

3) The authors wrote in the abstract, that senolytics and immunization against senescent cells is discussed in the manuscript. However, these points are rather listed than discussed. The Reviewer is missing a discussion of the advantages and possible risks, especially since some targets (such as ICAM-1 or NKG2D) have crucial functions. The authors should either include such discussion or moderate the statement (e.g. targets are listed).

4) 2.3 SASP: The authors should explain the background of immunosenescence and inflammaging. In this section it seems a bit controversial that pro-inflammatory cytokines are increasingly produced although immune cell activity is weakened. The authors are asked to explain here the role of innate immune cells to avoid misunderstandings (see Tulop et al. PMID 29375577).   

Quality of English is satisfying. However, the authors are asked to improove readibility as described in the Comments/Suggestions to the authors.

Reviewer 2 Report

This review article summarizes the existing literature regarding the immune system and cellular senescence.

Nevertheless, althought this topic has not previously reviewed, there is no solid literature regarding cellular senescence of immune cells. There is the assumption that immune cell types can enter senescence but, to my knowledge, there is no solid literature for this topic. 

Indeed, the main but basic issue of this review article is that it is more focused on explaining what senescence is (and it is really very well explained and summarisied). Despite of that, it cites almost no article regarding senescent immune cells (for example section "2.1. Morphological changes" or "2.6. Metabolic changes/disorders" do not touch at all immune cells, and many other sections are also general), because really there are no articles.

Some of them do not really go deeply into the topic or do not really establish the linke between cellular senescence and immune cell function: for example this one: "it has been found that during replicative aging, the decrease in CD28 expression is more pronounced in a population of CD8+ T lymphocytes (CD8+ CTLs) in comparison with 109 CD4+ T cells (CD4+ Ths) [39]." or ". This 113 may be why no significant number of senescent cells were found among CD28+ T cells". So they mainly cite not directly related results or negative results.

Sometimes there is a little bit of confusion between cellular senescence and ageing, which is expected, since they are closely related, for example "Тhe highest and most stable expression of p16 is observed in CD3+ T-lymphocyte from elderly donors".

I would suggest the authors to add other topics to the review article, since unfortunately there is not enough literature regarging immune cells and cellular senescence, and to modify the title accordingly.

Regarding the scientific quality, the article is well-written and not misleading, except the issues previously mentioned.

Round 2

Reviewer 2 Report

My issues have been addressed with comments, although the text has not been changed.